# Broad-Spectrum Antimicrobial Activity of Ultrafine (BiO)_2_CO_3_ NPs Functionalized with PVP That Can Overcome the Resistance to Ciprofloxacin, AgNPs and Meropenem in *Pseudomonas aeruginosa*

**DOI:** 10.3390/antibiotics12040753

**Published:** 2023-04-14

**Authors:** Bishnu D. Pant, Nalin Abeydeera, Rabindra Dubadi, Min-Ho Kim, Songping D. Huang

**Affiliations:** 1Department of Chemistry and Biochemistry, Kent State University, Kent, OH 44240, USA; bpant@kent.edu (B.D.P.); nkekiriw@kent.edu (N.A.); rdubadi@kent.edu (R.D.); 2Department of Biological Sciences, Kent State University, Kent, OH 44240, USA

**Keywords:** Bi-based antibacterial nanoparticles, ESKAPE pathogens, top-down synthesis of nanomaterials, antimicrobial resistance

## Abstract

Although it has no known biochemical role in living organisms, bismuth has been used to treat syphilis, diarrhea, gastritis and colitis for almost a century due to its nontoxic nature to mammalian cells. When prepared via a top-down sonication route from a bulk sample, bismuth subcarbonate (BiO)_2_CO_3_ nanoparticles (NPs) with an average size of 5.35 ± 0.82 nm exhibit broad-spectrum potent antibacterial activity against both the gram-positive and gram-negative bacteria including methicillin-susceptible *Staphylococcus aureus* (DSSA), methicillin-resistant *Staphylococcus aureus* (MRSA), drug-susceptible *Pseudomonas aeruginosa* (DSPA) and multidrug-resistant *Pseudomonas aeruginosa* (DRPA). Specifically, the minimum inhibitory concentrations (MICs) are 2.0 µg/mL against DSSA and MRSA and 0.75 µg/mL against DSPA and DRPA. In sharp contrast to ciprofloxacin, AgNPs and meropenem, (BiO)_2_CO_3_ NPs show no sign of developing Bi-resistant phenotypes after 30 consecutive passages. On the other hand, such NPs can readily overcome the resistance to ciprofloxacin, AgNPs and meropenem in DSPA. Finally, the combination of (BiO)_2_CO_3_ NPs and meropenem shows a synergistic effect with the fractional inhibitory concentration (FIC) index of 0.45.

## 1. Introduction

Bismuth subcarbonate (BiO)_2_CO_3_, the active ingredient in the milk of bismuth, has been a popular panacea for treating gastrointestinal disorders since the 1930s [1]. Except for its antimicrobial activity against *H. pylori* (or *C. pylori* as the strain was previously referred to) [2,3], little is known about its potential as a metal-based antimicrobial agent against other more dangerous pathogenic bacteria, namely *Staphylococcus aureus* and *Pseudomonas aeruginosa—*both belonging to the group of ESKAPE pathogens. The latter are a group of six highly virulent and multidrug-resistant opportunistic human pathogens consisting of *Enterococcus faecium*, *Staphylococcus aureus*, *Klebsiella pneumoniae*, *Acinetobacter baumannii*, *Pseudomonas aeruginosa*, and *Enterobacter species*. The paucity of reported antimicrobial studies on (BiO)_2_CO_3_ using the liquid microbiological culture technique is probably a result of its poor solubility in water or other organic solvents due to its infinite solid-state structure in which the (BiO)_n_ layers are sewed up by the chain of CO_3_^2−^ anions. It should be noted that water solubility, and hence absorption, is not necessary for the bulk (BiO)_2_CO_3_ to be employed in the therapy of conditions affecting the digestive tract as the site of action is within the gastrointestinal lumen. However, poor water or lipid solubility of this solid-state compound poses a limitation for its use in the treatment of the systemic or even topical infections. Inorganic nanoparticles (NPs) surface-coated with biocompatible and water-soluble polymer shells represent an attractive alternative to both topical and parenteral delivery of insoluble compounds. We have set out to investigate a variety of metal compounds as topical antimicrobial agents to address the unmet clinical need for new topical antimicrobial drugs to treat skin and soft tissue infections (SSTIs) [4,5,6,7,8]. Currently, there are only five topical antibiotics in clinical use for treating SSTIs. These include bacitracin, neomycin, polymyxin B, mupirocin and fusidic acid (i.e., fusidate), but neither of them is a broad-spectrum antibiotic. To improve the therapeutic efficacy against both gram-positive and gram-negative bacterial infections, the common approach is to combine three of the above topical antibiotics, i.e., bacitracin (active against gram-positive bacteria), neomycin (active against gram-negative bacteria) and polymyxin B (active against gram-negative bacteria) in a single ointment. However, this practice increases the chance of allergy development and/or other inadvertent side effects. Alternatively, a broad-spectrum oral or injectable antibiotic (e.g., ciprofloxacin) is often prescribed in combination with the chosen topical antibiotic to treat SSTIs. Both treatment regimens raise concerns about the overuse and misuse of antibiotics. An alternative strategy for treating SSTIs is the use of AgNPs as a topical antimicrobial agent, which has a venerable history and is gaining popularity in wound dressings, surface-coatings of medical devices, and even common household items [9,10,11]. AgNPs exhibit impressive antimicrobial activity against gram-negative bacteria with typical MIC values in the range of 1 μg/mL to 5 μg/mL [12], but AgNPs are much less effective against gram-positive bacteria with typical MIC values in the range of 1800 μg/mL to 2700 μg/mL [13]. Additionally, the unregulated wide use of AgNPs in consumer goods has now exposed an increasing number of bacterial strains to this treatment, causing Ag resistance [14].

In the era of rising antimicrobial resistance (AMR), it may be proven desirable to explore other metal compounds, such as broad-spectrum topical antimicrobial agents, as alternatives to conventional topical antibiotics for combating the serious threat of AMR, particularly in SSTIs. The antimicrobial modes of metal compounds are often very different from those of conventional antibiotics. The nonoverlapping mechanisms of metal compounds with those of conventional antibiotics sometimes render the broad-spectrum action possible, as well as the potential to defeat AMR in the mutant strains of bacteria. Within the scope of this paper, we concentrate on the synthesis, characterization, and antimicrobial properties of highly water-dispersible (BiO)_2_CO_3_ NPs that are surface-coated with biocompatible polyvinylpyrrolidone (PVP). The main purpose of this study is to explore the different antimicrobial mechanisms possessed by (BiO)_2_CO_3_ NPs in the hope to overcome the multi-drug resistance existing in various phenotypes of pathogenic bacteria. We show that the PVP-coated (BiO)_2_CO_3_ NPs exhibit potent broad-spectrum antimicrobial activity against both gram-positive and gram-negative bacteria including methicillin-susceptible *Staphylococcus aureus* (DSSA), methicillin-resistant *Staphylococcus aureus* (MRSA), drug-susceptible *Pseudomonas aeruginosa* (DSPA) and multidrug-resistant *Pseudomonas aeruginosa* (DRPA). We demonstrate that such (BiO)_2_CO_3_ NPs show no sign of developing Bi-resistant phenotypes after 30 passages, which is in sharp contrast to ciprofloxacin, AgNPs and meropenem. Furthermore, (BiO)_2_CO_3_ NPs can readily overcome the resistance to ciprofloxacin, AgNPs and meropenem developed in the DSPA mutant bacteria. On the other hand, the combination of (BiO)_2_CO_3_ NPs and meropenem shows a synergistic effect. Overall, (BiO)_2_CO_3_ NPs have potential as a broad-spectrum topical antimicrobial agent in place of conventional topical antibiotics, as a monotherapy or as combination therapies for treating SSTIs.

## 2. Results and Discussion

The bulk (BiO)_2_CO_3_ sample was first obtained as a colorless precipitate using a typical solid-state sol–gel reaction between bismuth nitrate and ammonium carbonate. The precipitate was then dehydrated at 200 °C for 4 h to afford the final product as a dry powder (Appendix A). To authenticate the phase and assess the purity of the bulk sample prepared using the sol–gel route, powder X-ray diffraction (PXRD) measurement on an as-synthesized bulk sample was performed, which confirms its phase identity and purity (Figure 1a). The diffraction peaks seen in the PXRD patterns were indexed into the tetragonal (BiO)_2_CO_3_ phase with the lattice constants of a = b = 3.865 Å and c = 13.675 Å (space group I_4_/mmm). These are in full agreement with those found on the reference card JCPDS No. 41-1488 [15]. Subsequently, (BiO)_2_CO_3_ NPs were produced by sonicating this phase-pure bulk sample for 18 h using polyvinylpyrrolidone (MW = 8000 g/mole, PVP-8k) as a surface modifier and dimethyl sulfoxide (DMSO) as a solvent [6]. Transmission electron microscopy (TEM) studies revealed that these NPs have an average size of 5.35 ± 0.82 nm (Figure 1b). The high-resolution transmission electron microscopy (HRTEM) measurements on individual NPs revealed a high degree of crystallinity in such NPs, with readily visible lattice spacings of 3.08 Å ± 0.3 Å (Figure 1c). 

After the successful synthesis of such ultrafine and water-dispersible (BiO)_2_CO_3_ NPs, we set out to test their antibacterial activity against a strain of drug-susceptible *Pseudomonas aeruginosa* (DSPA; ATCC 15692) and a strain of multidrug-resistant *Pseudomonas aeruginosa* (DRPA; ATCC BAA 2108)—gram-negative bacteria that are known to readily generate and transmit carbapenem-resistant genes [16,17,18]. For the purpose of comparing resistance development profiles between (BiO)_2_CO_3_ NPs and AgNPs, an Ag-resistant DSPA phenotype was generated by passing Ag-susceptible DSPA over 30 days. The minimum inhibitory concentration (MIC) was found to be 0.75 µg/mL against both DSPA (ATCC 15692) and DRPA (ATCC BAA 2108), after incubating the bacteria with (BiO)_2_CO_3_ NPs for 18 h (Table 1 and Appendix A). By comparison, AgNPs, which are considered to be one of the most effective metal-based topical antibacterial agents in PA bacteria, have a MIC of 1.69 µg/mL against DSPA. We then investigated whether (BiO)_2_CO_3_ NPs were active against a strain of methicillin-susceptible *Staphylococcus aureus* (MSSA; ATCC 6538) and a strain of methicillin-resistant *Staphylococcus aureus* (MRSA; ATCC BAA 44)—gram-positive bacteria that are prone to developing mupirocin and fusidate resistance. Incidentally, we found that AgNPs are completely ineffective against SA bacteria in our studies (MIC > 900 µg/mL) [6,19], while none of the earlier studies have provided evidence that (BiO)_2_CO_3_ NPs could be active against any gram-positive bacteria [20,21]. Our results showed that (BiO)_2_CO_3_ NPs exhibit a MIC of 2.0 µg/mL against both DSSA and MRSA (Table 1 and Appendix A), confirming that (BiO)_2_CO_3_ NPs possess broad-spectrum antibacterial activity. Overall, the antimicrobial activity of (BiO)_2_CO_3_ NPs is similar to that of AgNPs against the gram-negative PA bacteria, while their antibacterial activity is substantially better than that of AgNPs against the gram-positive SA bacteria.

To determine the required concentration of (BiO)_2_CO_3_ NPs to eradicate DSPA and DSSA, we measured colony-forming units (CFUs) at varying concentrations of NPs using the conventional plating method in reference to AgNPs and Bi(NO_3_)_3_. At the concentration of 1.5 µg/mL, (BiO)_2_CO_3_ NPs could completely eradicate DSPA bacteria with a 6 − log reduction in CFUs, while at the same concentration, AgNPs could only achieve approximately a 1 − log reduction in CFUs. It is worth noting that Bi(NO_3_)_3_, an ionic salt of Bi, was shown to be inactive against DSPA bacteria (Figure 2a and Appendix A). Similarly, the CFU measurements in DSSA revealed a similar situation (Figure 2b). Specifically, at the concentration of 3.0 µg/mL, (BiO)_2_CO_3_ NPs caused the growth reduction of DSSA bacteria by a 3.5 − log of CFUs, while at the same concentration, Bi(NO_3_)_3_ only caused a slight reduction in CFUs (<<1 − log). It should be noted that AgNPs were not included in the latter studies as a comparison group because the previous studies repeatedly showed that AgNPs were not active against SA bacteria [6,13].

Encouraged by the potent broad-spectrum antimicrobial activity, we proceeded to examine the resistance development profile of (BiO)_2_CO_3_ NPs in comparison with those of ciprofloxacin, AgNPs and meropenem in DSPA bacteria. For the first 30 passages, cultures of DSPA were treated independently with (BiO)_2_CO_3_ NPs, ciprofloxacin, AgNPs and meropenem, respectively, each at a sub-lethal dose. After 13 days, the MIC for ciprofloxacin increased by 32-fold, suggesting that DSPA bacteria were now ciprofloxacin resistant. Likewise, after 12 days, AgNPs (MIC = 54 µg/mL or 32 × MIC_0_) were no longer effective, and the MIC climbed to 64 × MIC_0_ after another 3 days, indicating that total resistance was developed by DSPA to AgNPs. Further, the MIC for meropenem showed a gradual increase after 5 days, and by day 15, the MIC increased by 64-fold, indicating the total resistance of DSPA to meropenem (Figure 2c). In stark contrast, (BiO)_2_CO_3_ NPs maintained their initial MIC of 0.75 µg/mL during all 30-day passages, indicating that (BiO)_2_CO_3_ NPs are not prone to developing Bi resistance in PA bacteria under such conditions. Given the outstanding performance of (BiO)_2_CO_3_ NPs on the delayed development of Bi-resistant phenotypes, we hypothesized that these NPs might overcome the resistance developed in the DSPA mutant bacteria that are resistant to ciprofloxacin, AgNPs and meropenem. The latter is an antibiotic in the carbapenem family and is used only as a last resort. To investigate this possibility, we first produced ciprofloxacin-, AgNPs- and meropenem-resistant mutants of DSPA bacteria referred to as PA^cipR^, PA^AgR^, and PA^meroR^, respectively, using the threshold of 32 × MIC_0_ as the limit to select a specific PA mutant for this study. Hence, all these mutants were obtained after 15-day passages, where the value of 32 × MIC_0_ was reached by the DSPA bacteria. Each of these DSPA mutants was then treated with (BiO)_2_CO_3_ NPs for an additional 30 days to detect any change in MIC values. The MIC of (BiO)_2_CO_3_ NPs against PA^cipR^ bacteria was 1.0 µg/mL, as shown in Figure 2d, and there was no sign of Bi resistance development for the next 29 days. Similarly, the MIC of (BiO)_2_CO_3_ NPs against PA^AgR^ bacteria was 1.25 µg/mL, and there was no sign of Bi resistance development for the next 29 days either. In contrast, the MIC of (BiO)_2_CO_3_ NPs against PA^meroR^ bacteria was 1.75 µg/mL but showed a slight increase to 2.5 µg/mL on day 8 and a further increase to 10 µg/mL on day 23, and finally 20 µg/mL on day 27, indicating a decreased susceptibility to Bi in these mutant bacteria (Appendix A). We attribute both the resilience to Bi resistance development and the ability of (BiO)_2_CO_3_ NPs to overcome the drug resistance in the mutant strains of PA^cipR^, PA^AgR^ and PA^meroR^ to the unique and exclusive mechanisms of action that are not shared by ciprofloxacin, AgNPs and meropenem, indicating that there is no cross resistance experienced by (BiO)_2_CO_3_ NPs.

Previous studies have revealed that both ROS generation and physical membrane disruption are key antimicrobial mechanisms of action of many nanomaterials, both of which are not common in conventional antibiotics [22,23]. First, time-kill experiments against DSPA were carried out using different concentrations of (BiO)_2_CO_3_ NPs to determine the time required for evaluating the cellular uptake of (BiO)_2_CO_3_ NPs, ROS generation and cell membrane disruption in bacterial cells. With the exception of the highest tested concentration of 1.6 × MIC (1.25 µg/mL) at which bacterial growth was suppressed after 6 h, (BiO)_2_CO_3_ NPs at the concentration of 1.3 × MIC (1.00 µg/mL) seemed to have a delayed lethal action, i.e., the bacterial growth was suppressed only after 9 h, while at the concentration of MIC (0.75 µg/mL), these NPs exhibited a bacteriostatic effect after 12 h (Appendix A). Hence, we selected 6 h as the treatment duration for the Bi-uptake assays. Subsequently, DSPA bacterial cells were first treated with varying concentrations of (BiO)_2_CO_3_ NPs and incubated for 6 h; the cells were then lysed with concentrated nitric acid. The quantification of Bi concentrations in the lysed cells was performed using flame atomic absorption spectroscopy (AAS). The outcomes showed that the bacterial cells took up considerable amounts of NPs, even at the concentrations substantially below the MIC (Figure 3a). As a result of these findings, we concluded that the observed delay in bacterial killing at concentrations lower than 1.3 × MIC was unlikely to be caused by the slow rate of cellular uptake of (BiO)_2_CO_3_ NPs. The intracellular levels of ROS in DSPA bacteria treated with varying concentrations of (BiO)_2_CO_3_ NPs were then assessed using the 2′, 7′-dichlorofluorescein diacetate (DCFHDA) cellular ROS assay. The results showed that the intracellular levels of ROS in DSPA were substantially enhanced by (BiO)_2_CO_3_ NPs in a dose-dependent manner when normalized with the cell population of living bacteria (Figure 3b), implying that intracellular ROS generation is at least partially responsible for bacterial cell death. However, the exact biochemical pathways of Bi-triggered ROS generation may be complex and remain unknown, given that Bi is known to react with a plethora of biomolecules including nucleosides, nucleotides, peptides, proteins and enzymes. 

To investigate whether (BiO)_2_CO_3_ NPs could physically disrupt the integrity of the bacterial membrane as another potential mode of action for bacterial killing, scanning electron microscopy (SEM) imaging experiments were performed using the sample preparation procedure reported previously [7]. The SEM images revealed that in the (BiO)_2_CO_3_ NP-treated DSPA cells, pits and kinks were formed on the external cell surfaces and there was an evident change in their cell morphology as opposed to the untreated control DSPA cells. In the DSSA cells treated with (BiO)_2_CO_3_ NP, the intercellular spaces were littered with a substantial amount of unidentified protein matrix as a result of leaked cytosol, with occasional fissures formed on the cell surfaces. Overall, our results indicate that the cell membrane function is impaired in both DSPA and DSSA (Figure 3c–f), suggesting that the damage and impairment of cell membrane integrity by (BiO)_2_CO_3_ NPs is possibly another cause of bacterial cell death [24].

To determine the therapeutic selectivity of (BiO)_2_CO_3_ NPs, their toxicity towards human dermal fibroblasts (HDFs) and murine macrophage-like cells (RAW 264.7) were tested. (BiO)_2_CO_3_ NPs were shown to be nontoxic against both HDF and RAW cells with half-maximal inhibitory concentrations (IC50 > 50 µg/mL for HDF and >40 µg/mL for RAW cells; see Appendix A). These findings suggest that (BiO)_2_CO_3_ NPs are more selective for bacteria than for healthy mammalian cells. The selectivity indexes (SI = $\frac{\mathrm{IC}50}{\mathrm{MIC}}$) of DSPA and DSSA bacteria over HDF and RAW cells are calculated and given in Table 2.

The demonstrated ability of (BiO)_2_CO_3_ NPs to overcome meropenem resistance prompted us to investigate whether (BiO)_2_CO_3_ NPs could have a synergistic or additive effect with meropenem, given that these two different antimicrobial agents are likely to act by noncoinciding mechanisms of action [25]. A checkerboard experiment with a 96-well plate was used to measure the degree of synergism. The MIC of (BiO)_2_CO_3_ NPs was found to be 0.75 µg/mL against DRPA, while the MIC of meropenem was found to be 2.0 µg/mL against the same strain of bacteria, as shown by the wells on the leftmost column and bottom row, respectively. To determine the fractional inhibitory concentration, mixtures of the two doses were created using the two-fold dilution procedure (FIC; Appendix A). The FIC index is used to determine if two therapies are synergistic (χ ≤ 0.5), additive (0.5 < χ < 4) or antagonistic (χ ≥ 4) when they are administered together [26]. The MIC_mero_ was considerably reduced from 2.0 µg/mL to 0.25 µg/mL against DRPA when meropenem was coupled with 1/3MIC of (BiO)_2_CO_3_ NPs (Figure 4), and the corresponding FIC index was 0.45. Such data strongly suggest that (BiO)_2_CO_3_ NPs and meropenem have a synergistic effect (Appendix A) resulting from the noncoinciding modes of action of (BiO)_2_CO_3_ NPs and meropenem, which point to the possibility of a combination therapy of topical plus parenteral delivery of the two for treating STTIs caused by persistent bacterial cells.

## 3. Conclusions

In conclusion, we have shown that (BiO)_2_CO_3_ NPs functionalized with PVP are not only an exceptionally potent broad-spectrum antibacterial agent but are also resilient to the development of Bi resistance. More importantly, (BiO)_2_CO_3_ NPs can maintain antimicrobial activity against ciprofloxacin-, AgNPs- and meropenem-resistant mutants of PA bacteria. As a result, (BiO)_2_CO_3_ NPs have potential as a broad-spectrum topical antimicrobial agent in place of antibiotics in monotherapy for treating polymicrobial and/or multidrug-resistant SSTIs. Additionally, the synergistic effect derived from the combination of (BiO)_2_CO_3_ NPs with meropenem shows the potential of using (BiO)_2_CO_3_ NPs in combination therapy for treating more severe STTIs. Given the fact that bismuth subcarbonate itself has been an over-the-counter medication for almost a century, our findings in this study demonstrate that a drug repurposing approach, coupled with novel synthetic procedures for preparing nanomaterials, could offer a very tiny answer to the huge problem of AMR.

## 4. Materials and Methods

### 4.1. Chemical Reagents and Biological Material

All chemical reagents were purchased from commercial sources and were not purified before use. Bismuth (III) pentahydrate (98%), nitric acid (65%), poly (vinylpyrrolidone) (*Mw* = 8000), dimethyl sulfoxide (≥99.5%), silver nitrate (≥99%), ammonium carbonate, D-maltose, ciprofloxacin (≥98%) and meropenem (≥98%) were purchased from Sigma-Aldrich. Bacterial strains, growth media and antibiotics, and gram-positive bacteria (SA; ATCC 6538, MRSA; ATCC BAA-44) and gram-negative bacteria (DSPA; ATCC 15692, DRPA; ATCC BAA-2108) were purchased from American Type Culture Collection. Tryptic broth powder (TSB), tryptic soy agar (TSA), nutrient broth (NB) and nutrient agar (NA) were purchased from Fisher Scientific.

### 4.2. Synthesis Methods

Synthesis of (BiO)_2_CO_3_: Bi(NO_3_)_3_ aqueous solution was prepared under acidic conditions of HNO_3_. A white precipitate of (BiO)_2_CO_3_ was formed by slowly adding 40 mL of ammonium carbonate solution while stirring continuously. This product went through three separate rounds of filtration and washing with deionized water. When the washing process was complete, the pH of the water was about 7.0. This white product was then heated on a furnace at 200 °C for 4 h to yield white colored bulk (BiO)_2_CO_3_.

Synthesis of (BiO)_2_CO_3_ NPs: Sonomechanically synthesized (BiO)_2_CO_3_ NPs were made from previously prepared bulk (BiO)_2_CO_3_. Then, 1 mg of finely milled bulk (BiO)_2_CO_3_ was disseminated in 1 mL of DMSO containing 100 mg of PVP-8k to accomplish this. After this, the mixture was sonicated for 18 h.

Synthesis of AgNPs: Through the use of a modified version of the Tollens method, AgNPs were produced by reducing [Ag(NH_3_)_2_]^+^ with D-maltose [27]. The particle shape and size were not measured but were assumed to be spherical with a diameter of 28 nm based on the results described in the original article. We measured the Ag concentration using atomic absorption spectroscopy (AAS).

### 4.3. Characterization 

Powder XRD: The powder pattern was acquired with the use of a Rigaku MiniFlex 600 X-ray diffractometer by employing Cu Kα radiation, a Kβ-filter, a LynxEye PSD detector and an incident beam Ge111 monochromator. Patterns were evaluated using a step size of 0.01446° degrees and an exposure time of 800 s, moving from 10 to 70° 2 theta degrees.

TEM and HRTEM: The following procedure was used to make TEM grids. The (BiO)_2_CO_3_ NPs were disseminated in ethanol for 30 min before being sonicated. The suspension was then dropped onto a carbon-coated copper TEM grid (400-mesh) and the samples were allowed to air-dry before being analyzed. Images were taken with an FEI Tecnai F20 (200 kV) TEM equipped with a field emission gun and a scanning TEM (STEM) instrument. FIJI was used to process TEM pictures [28].

SEM imaging of bacteria: To investigate the morphology of bacterial samples, a Quanta 450 scanning electron microscope equipped with a 15 kV accelerating voltage was utilized. Before being examined, the bacteria of interest (1 × 10^9^ CFU/mL) were routinely subjected to treatment with nanoparticles at varying concentrations and for varying amounts of time. Before being subjected to tannic acid at a concentration of one percent, pellets were washed with PBS three times. The samples were then dried in the air, dehydrated using a succession of graded ethanol solutions, gold-coated, and, finally, scanning electron micrographs were obtained of the samples.

### 4.4. Biological Assays

Preparation of Test Solutions: The necessary amounts of (BiO)_2_CO_3_ NPs and PVP as the negative controls (by weight) were dispersed in DMSO to make test solutions. The powder was used to make nutrient media for microorganisms (TSB, TSA, NB, and NA) by dissolving the necessary amount in DI water.

Bacteria Suspensions: In order to culture the bacterial suspensions, an isolated colony of gram-positive bacteria was added to 5 mL of TSB media and a colony of gram-negative bacteria was added to 5 mL of NB media. The bacterial suspensions were then placed in an incubator at 37 °C for 18 h. To quantify the cell density of bacterial cells, a SpectraMax M4 Microplate Reader was used to take measurements of the optical density at 600 nm.

MIC assay: After diluting colloidal nanoparticle solutions of varying concentrations in TSB, they were injected onto a 96-well plate with a strain of bacteria at a concentration of 10^6^ CFU/mL. This was followed by an incubation period of 18 h at 37 degrees Celsius. After this, the lowest concentration of NPs that suppresses observable development of the investigated microbes, with unassisted sight and an OD measurement at 600 nm using a microtiter plate reader, was identified.

Colony Forming Unit (CFU/mL) Assays: Bacteria grown in TSB medium without nanoparticles would serve as a control in all cell culture experiments. After 18 h of incubation, a diluted mixture containing the nanoparticles was disseminated across agar plates using glass spreaders. Later on, the colonies formed in each plate were quantified and converted to CFU/mL values. Triplicate readings were obtained for all technical and biological parameters. By visual reading and OD at 600 nm using a microtiter plate reader, the MIC was determined as the lowest dose of a drug that may inhibit the growth of a microbe.

Bi uptake by PA measured with AAS: A flame AAS was used to measure Bismuth cellular absorption in DSPA (Buck Scientific Atomic Absorption Spectrometer, Model 210 VGP). In order to create the calibration curve, an available commercial standard solution curve (1000 ppm) was diluted to a variety of concentrations. Metal concentrations were determined using a hollow cathode lamp operation. For the process, different (BiO)_2_CO_3_ concentrations were utilized. Six hours after being incubated at 37 °C, a 500 µL portion of the bacterial suspension was taken, and the CFU count was calculated using the agar plate method and centrifugation was performed for the remaining bacterial cultures to obtain the bacterial pellets. To remove the organic matter, 70 percent HNO_3_ was added to the pellet before calcination at 620 degrees Celsius for five hours, which turned the metal ions in the solutions into oxides. Dissolving these oxides in aqua regia allowed for the measurement of bismuth content.

Quantification of reactive oxygen species: DCFH-DA assay was used to measure the amount of intracellular ROS generated by (BiO)_2_CO_3_ nanoparticles in DSPA. After first centrifuging 1 milliliter of DSPA that had been cultivated overnight, the pellet was then resuspended in new NB media. The bacteria were then subjected to an incubation with varying doses of (BiO)_2_CO_3_ NPs in comparison with a control for a period of sixty minutes. After that, the cells from each group were collected by centrifugation, and then they were washed using PBS. The microbial cells were then mixed with 500 μL of DCFH-DA dye and incubated for an additional half hour. 

The bacterial cells were incubated with 500 μL of 20 μM DCFH-DA dye in PBS at 37 °C while shaking for 30 min. The level of reactive oxygen species found inside of cells was measured using fluorescence microscopy, with the excitation and emission wavelengths adjusted at 497 nm and 529 nm, respectively.

Drug resistance study: The MIC was measured using the dilution method. Bacteria at a concentration of (~1.0 × 10^6^ CFU/mL) were grown in an NB medium supplemented either with (BiO)_2_CO_3_ NPs, ciprofloxacin, AgNPs, or meropenem. After preparation, the bacterial solutions were kept in an incubator at 18 h. After the incubation period, the MIC was calculated as the lowest concentration that prevented the microorganisms being tested from visibly growing. The bacteria were cultured in a serial fashion until they reached 30 days, at which point the operation was terminated. After that, three distinct forms of mutant PA were created by subjecting the original strain to a 15-day treatment with AgNPs, ciprofloxacin, and meropenem. Following this step, mutant bacteria were taken and then treated with (BiO)_2_CO_3_ NPs for an additional thirty days.

Mammalian cell viability assay: An MTT viability experiment was utilized to measure the level of cytotoxicity that (BiO)_2_CO_3_ nanoparticles exhibited against mammalian cells. In a 96-well plate, mammalian cells (RAW 264.7 cells and normal human dermal fibroblasts) were seeded at a density of 4 × 10^4^ cells per well with a DMEM high-glucose medium. The plate was then placed in an incubator for 24 h. After that, 100 μL of media with various test doses of (BiO)_2_CO_3_ NPs were added to each well and allowed to incubate for an additional 24 h. Following the incubation of the cells with 10 μL of the MTT reagent for 2 h, 100 μL of the detergent reagent was added to each well, and the plate was then left covered for another 2 h. With the use of a microplate reader, the absorbance was determined at 570 nm. The experiment was carried out three times, and the findings were reported as a percentage of viable cells. 

Checkerboard assay: As previously mentioned, a checkerboard assay was performed in a 96-well plate [25]. After serially diluting meropenem by a factor of 2 along the row axis and serially diluting (BiO)_2_CO_3_ NPs by a factor of 2 down the column axis, the resulting matrix had a mixture of (BiO)_2_CO_3_ NPs and meropenem at varying concentrations in each well. To calculate the MIC, every well was infected with DRPA at a concentration of about 1 × 10^6^ CFU/mL in a final volume of 100 μL. After being incubated for 18 h, the MIC was inspected for visibility. The FIC of (BiO)_2_CO_3_ NPs is computed by taking the MIC of (BiO)_2_CO_3_ NPs when they are exposed to the antibiotic and dividing that number by the MIC of (BiO)_2_CO_3_ NPs when they are not exposed to the antibiotic. In a similar manner, the FIC of an antibiotic was determined by dividing the antibiotic’s MIC, while it was in the presence of (BiO)_2_CO_3_ NPs, by the antibiotic’s MIC in the absence of (BiO)_2_CO_3_ NPs. The FIC index can therefore be interpreted as synergistic (χ ≤ 0.5), additive (0.5 < χ < 4), or antagonistic (χ ≥ 4) based on the sum of both FIC values.

Statistical analysis: Statistical analysis was performed using GraphPad Prism version 8.0 software. A two-tailed unpaired *t*-test was used to determine statistical significance between two groups. A statistical significance among multiple groups was analyzed using one-way ANOVA followed by the Holm–Sidak comparisons test. For all analyses, a *p*-value of less than 0.05 was considered to be statistically significant. Data were presented as mean ± standard deviation (mean ± s.d). The in vitro studies were run with at least three biological replicates, and each biological replicate had three technical replicates.

## Figures and Tables

**Figure 1 antibiotics-12-00753-f001:**
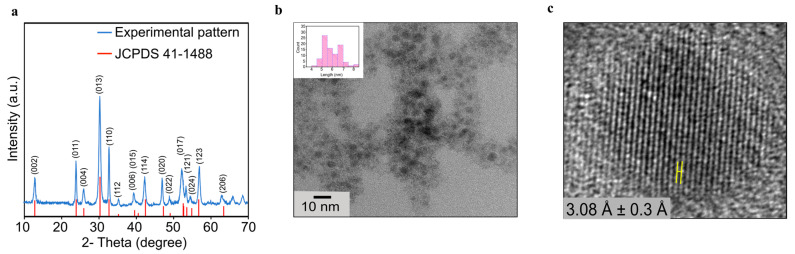
Characterization of (BiO)_2_CO_3_ samples. (**a**) Powder XRD patterns of a bulk sample in comparison with the simulated patterns; (**b**) a representative TEM image of BiO)_2_CO_3_ NPs with the size distribution histogram as inset; and (**c**) an HRTEM image of a representative single NP.

**Figure 2 antibiotics-12-00753-f002:**
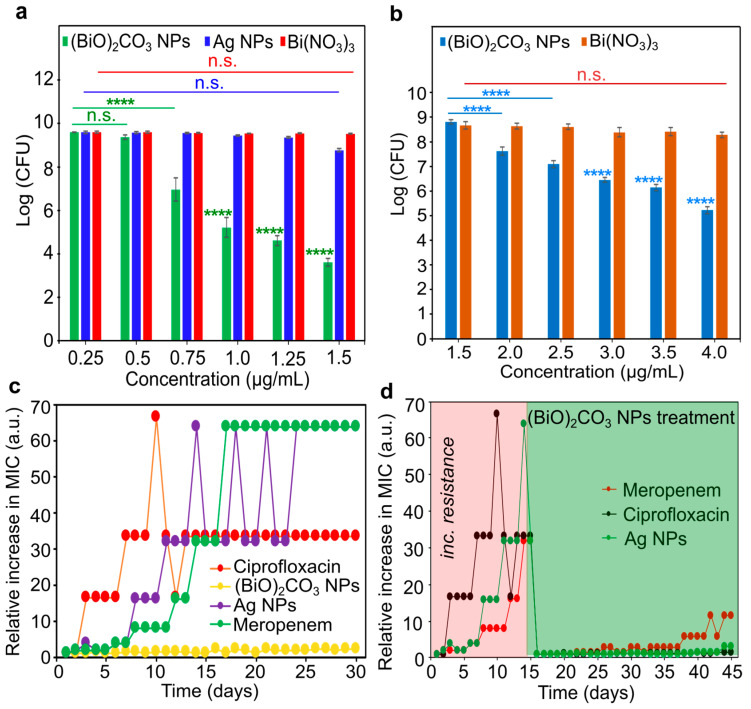
Antibacterial activity studies. (**a**) growth inhibitory curves of (BiO)_2_CO_3_ NPs against DSPA vs. Bi(NO_3_)_3_ and AgNPs; (**b**) growth inhibitory curves of (BiO)_2_CO_3_ NPs against DSSA vs. Bi(NO_3_)_3_; and (**c**) profiles of resistance development of (BiO)_2_CO_3_ NPs vs. ciprofloxacin, AgNPs and meropenem in DSPA after 30 days of treatment with each drug; and (**d**) declined drug resistance by the treatment of (BiO)_2_CO_3_ NPs in the DSPA bacteria that are resistant to ciprofloxacin, AgNPs and meropenem. Not significant (n.s.), and ****** (*p* < 0.0001).

**Figure 3 antibiotics-12-00753-f003:**
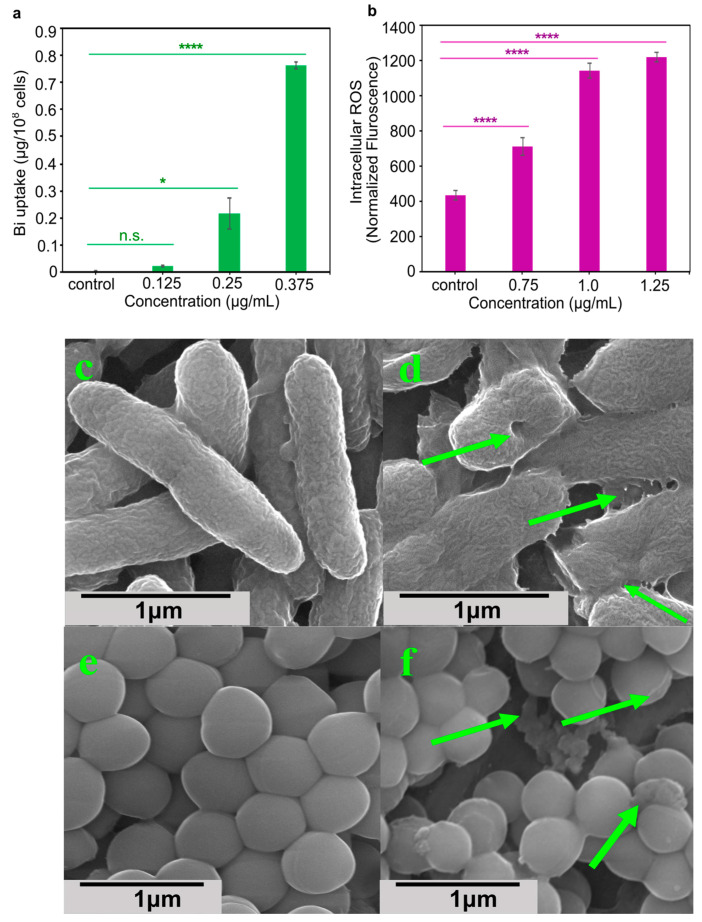
Antimicrobial mechanistic studies of (BiO)_2_CO_3_ NPs. (**a**) Cellular uptake of (BiO)_2_CO_3_ NPs in DSPA bacteria as measured by Bi concentrations in the cell lysates after 6 h of incubation; (**b**) standardization of ROS production following treatment of DSPA bacterial cells with three doses of (BiO)_2_CO_3_ NPs; (**c**) SEM image of the control DSPA cells in comparison with the (**d**) DSPA cells reacted with (BiO)_2_CO_3_ NPs; and/or (**e**) SEM image of the untreated control SA cells in comparison with (**f**) DSSA cells reacted with (BiO)_2_CO_3_ NPs. Not significant (n.s.), *** (*p* < 0.05), and ****** (*p* < 0.0001).

**Figure 4 antibiotics-12-00753-f004:**
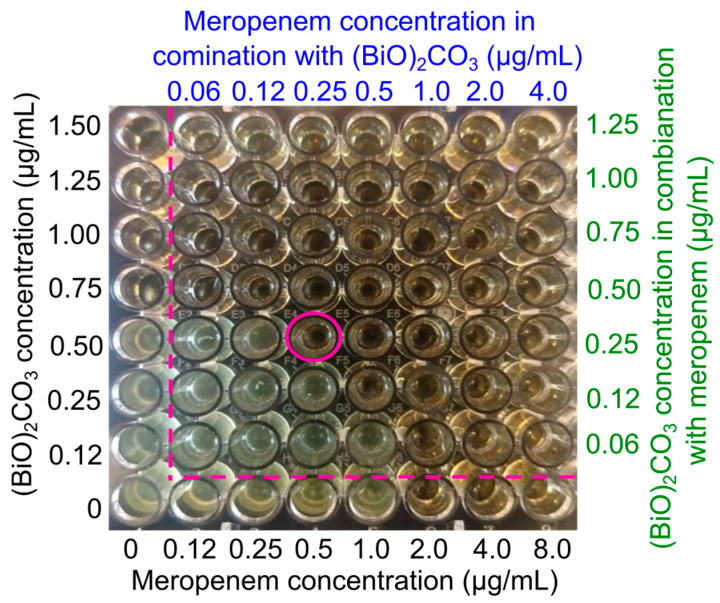
Checkerboard assay for (BiO)_2_CO_3_ NPs with meropenem against DSPA.

**Table 1 antibiotics-12-00753-t001:** MIC values of (BiO)_2_CO_3_ NPs against four different strains of bacteria.

Bacteria	MIC (µg/mL)
Drug-susceptible *Pseudomonas aeruginosa* (ATCC 15692)	0.75
Multidrug-resistant *Pseudomonas aeruginosa* (ATCC BAA 2108)	0.75
Methicillin-susceptible *Staphylococcus aureus* (ATCC 6538)	2.0
Methicillin-resistant *Staphylococcus aureus* (ATCC BAA-44)	2.0

**Table 2 antibiotics-12-00753-t002:** Selectivity index (SI =$\;\frac{\mathrm{IC}50}{\mathrm{MIC}}$) of DSPA and DSSA bacteria over HDF and macrophage-like RAW 264.7 cells.

Species	SI (HDF)	SI (RAW 264.7 Cells)
DSPA	>80	>53
DSSA	>30	>20

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
