# Peer review of "Broad-Spectrum Antimicrobial Activity of Ultrafine (BiO)2CO3 NPs Functionalized with PVP That Can Overcome the Resistance to Ciprofloxacin, AgNPs and Meropenem in Pseudomonas aeruginosa"

_antibiotics, 2023, doi:10.3390/antibiotics12040753_

Round 1
Reviewer 1 Report
Comments to the Authors
This article titled “Broad-spectrum antimicrobial activity of ultrafine (BiO)2CO3 NPs that can overcome the resistance of ciprofloxacin, AgNPs and meropenem in P. aeruginosa.
The language and arrangement of article is very good. However, What is the main meanings of this study? Add them in the introduction. Thus, this article can be accepted after major revision.
1. Revise the format of (BiO)2CO3 in the whole article.
2. Check the experimental design, is it reasonable?, author should evaluate the resistance in vivo (cell level). It is good if authors can add the proof and resistance results of ciprofloxacin, AgNPs and meropenem at animal level.
Author Response
This article titled “Broad-spectrum antimicrobial activity of ultrafine (BiO)2CO3 NPs that can overcome the resistance of ciprofloxacin, AgNPs and meropenem in P. aeruginosa.
The language and arrangement of article is very good. However, What is the main meanings of this study? Add them in the introduction. Thus, this article can be accepted after major revision.
Answer: A short statement: “The main purpose of this study is to explore the different antimicrobial mechanisms possessed by (BiO)2CO3 NPs in the hope to overcome multi-drug resistance existing in various phenotypes of pathogenic bacteria.” is added to the introduction to spell out the main meanings of this study.
- Revise the format of (BiO)2CO3 in the whole article.
Answer: We have checked the formula (BiO)2CO3 throughout the entire article to make sure they are correct and consistent.
2. Check the experimental design, is it reasonable?, author should evaluate the resistance in vivo (cell level). It is good if authors can add the proof and resistance results of ciprofloxacin, AgNPs and meropenem at animal level.
Answer: Because of the potent in vitro activity of (BiO)2CO3 NPs, we have plans to conduct in vivo animal test of these NPs in comparison with ciprofloxacin, AgNPs and meropenem. However, the results won’t be available to be included in this manuscript for publication due to the time restriction.
Reviewer 2 Report
Dear Autors,
The curent work presents complex results, supported by well-grounded arguments. It's true, we have a huge problem of AMR, but your findings in this study demonstrate that drug repurposing approach, coupled with novel synthetic procedures for preparing nanomaterials, could be a solution.
The bibliography is relatively current and is in agreement with research on (BiO)2CO3 nanoparticles with potential for use as a broad-spectrum topical antimicrobial agent in place of conventional topical antibiotics as monotherapy or combination therapies to treat SSTIs.
The work can be published after the elimination of small drafting errors.
Author Response
We'd like to thank this reviewer for the positive comments on this manuscript. Since there is no specific issue(s) raised by the reviewer, we do not have any point-by-point answers given here.
Reviewer 3 Report
The manuscript “Broad-spectrum antimicrobial activity of ultrafine (BiO)2CO3 NPs that can overcome the resistance of ciprofloxacin, AgNPs and meropenem in P. aeruginosa” aimed to synthesize (BiO)2CO3 NPs, functionalize them, and analyze their antimicrobial potential against P. aeruginosa resistant to ciprofloxacin, AgNPs and meropenem.
The title needs to be modified to inform readers that the nanomaterial assayed was not (BiO)2CO3 NPs, but their functionalized version, obtained using PVP. P. aeruginosa needs to be written without abbreviation as it is the first appearance in the manuscript and also in italics. The numbers 2 and 3 in the formula of nanomaterial need to be written subscript.
In “Introduction” the concept of “ESKAPE pathogens” should be explained to readers.
When it comes to “Results and Discussion”, in line 93 “so-gel” needs to be corrected to “sol-gel”. It is also necessary to remove the method’s description (f.ex. “The precipitate was then dehydrated at 200ºC for 4 hours to afford the final product as dry powder”) from this section and place it in the correct section: “Methods”. No results related to the characterization of AgNPs synthesized by the authors were presented. It is not possible to know their shape, surface charge, chemical constitution, etc. It is necessary to present these results.
Figures 2 and 3 (a and b) lack statists: asterisks or letters signaling the results from statistical tests for the reader to visualize which result was a significant one.It needs to be presented.
Regarding “Methods”, there is no section to describe statistics, and it is essential to provide it and perform the tests. Characterization of AgNPs is also missing. Tests of toxicity using PVP alone need to be performed and their results, presented in Figure 2a.
When it comes to “Conclusion”, it needs to refer to the nanomaterial as a functionalized one. It is very important to highlight that. The study used a version of (BiO)2CO3 modified by PVP. The lack of statistics impairs that readers can understand which result is a significant one. So, it is not possible to consider the “Conclusion” as supported by the data presented.
English language and style are fine; minor spell check is required.
Author Response
Comments to the Authors
The manuscript “Broad-spectrum antimicrobial activity of ultrafine (BiO)2CO3 NPs that can overcome the resistance of ciprofloxacin, AgNPs and meropenem in P. aeruginosa” aimed to synthesize (BiO)2CO3 NPs, functionalize them, and analyze their antimicrobial potential against P. aeruginosa resistant to ciprofloxacin, AgNPs and meropenem.
The title needs to be modified to inform readers that the nanomaterial assayed was not (BiO)2CO3 NPs, but their functionalized version, obtained using PVP. P. aeruginosa needs to be written without abbreviation as it is the first appearance in the manuscript and also in italics. The numbers 2 and 3 in the formula of nanomaterial need to be written subscript.
Answer: We thank this reviewer for the invaluable suggestion about the title and have changed it into:” Broad-spectrum antimicrobial activity of ultrafine (BiO)2CO3 NPs functionalized with PVP and can overcome the resistance of ciprofloxacin, AgNPs and meropenem in Pseudomonas aeruginosa”.
In “Introduction” the concept of “ESKAPE pathogens” should be explained to readers.
Answer: We have added a short explanation: “The latter are a group of six highly virulent and multidrug-resistant opportunistic human pathogens consisting of Enterococcus faecium, Staphylococcus aureus, Klebsiella pneumoniae, Acinetobacter baumannii, Pseudomonas aeruginosa, and Enterobacter species” after the phrase containing ESKAPE pathogens
When it comes to “Results and Discussion”, in line 93 “so-gel” needs to be corrected to “sol-gel”. It is also necessary to remove the method’s description (f.ex. “The precipitate was then dehydrated at 200ºC for 4 hours to afford the final product as dry powder”) from this section and place it in the correct section: “Methods”. No results related to the characterization of AgNPs synthesized by the authors were presented. It is not possible to know their shape, surface charge, chemical constitution, etc. It is necessary to present these results.
Answer: Again, this reviewer has a very good point on suggestion to rearrange the paragraphs. We have followed this suggestion and made the appropriate changes. With regard to the synthesis and characterization of AgNPs, the following paragraph is added in the Method:
Synthesis of AgNPs: Through the use of a modified version of the Tollens method, AgNPs were produced by reducing [Ag(NH3)2]+ with D-maltose.[27] The particle shape and size were not measured but assumed to be spherical with a diameter of 28 nm based on the results described into the original article. We measured the Ag concentration using atomic absorption spectroscopy (AAS).
Figures 2 and 3 (a and b) lack statists: asterisks or letters signaling the results from statistical tests for the reader to visualize which result was a significant one. It needs to be presented.
Answer: We have conducted statistical analyses for the results when they are deemed necessary. Particularly, Figures 2 and 3 (a and b) are replaced with new graphs reporting such results.
Regarding “Methods”, there is no section to describe statistics, and it is essential to provide it and perform the tests. Characterization of AgNPs is also missing. Tests of toxicity using PVP alone need to be performed and their results, presented in Figure 2a.
Answer: First, a section on Statistical Analysis was added to Method. Second, PVP was tested to be nontoxic to cells. Consequently, we used an appropriate amount of PVP in the cell culture medium to stabilizing the NPs. Second, a statement in the Method section was added: “Preparation of Test Solutions: The necessary amount of (BiO)2CO3 NPs and PVP as the negative control (by weight) were dispersed in DMSO to make test solutions.” Third, synthesis and characterization of AgNPs have been reported in the literature for multiple times. In this study, AgNPs were merely used as a reference material. We followed a literature method for preparing such NPs and assume that our NPs have similar shape and size as reported in the literature. Please see the above to the extra paragraph to added to Method section to describe their synthesis.
When it comes to “Conclusion”, it needs to refer to the nanomaterial as a functionalized one. It is very important to highlight that. The study used a version of (BiO)2CO3 modified by PVP. The lack of statistics impairs that readers can understand which result is a significant one. So, it is not possible to consider the “Conclusion” as supported by the data presented.
Answer: Once again, we added “functionalized with PVP” after (BiO)2CO3 NPs in Conclusion section to emphasize the potential role played by PVP. After statistical analysis of our results in Figures 2 and 3 (a and b), we are confident that our conclusions are firmly supported by the experimental data presented in the manuscript as judged by the P values.
Round 2
Reviewer 1 Report
ok
Reviewer 3 Report
Manuscript improved.